# Characterization of Non-Gaussianity in the Snow Distributions of Various Landscapes

**Noriaki Ohara[1], Andrew D. Parsekian[1], Benjamin M. Jones[2], Rodrigo C. Rangel[1], Kenneth M. Hinkel[3], and Rui A. P. Perdigão[4, 5]**

[1]University of Wyoming, Laramie, WY, USA

[2]University of Alaska Fairbanks, Fairbanks, AK, USA

[3]Michigan Technological University, Houghton, MI, USA

[4]Meteoceanics Institute for Complex System Science, IUC Physics of Complex Coevolutionary Systems & Fluid Dynamical Systems, Washington, DC, USA

[5]Synergistic Manifolds, Lisbon, Portugal

Corresponding author: Noriaki Ohara (nohara1@uwyo.edu)

**Summary**

Snow distribution characterization is essential for accurate snow water estimation for water resource prediction from existing in-situ observations and remote sensing data at a finite spatial resolution. Four different observed snow distribution datasets were analyzed for Gaussianity. It was found non-Gaussianity of snow distribution is a signature of wind redistribution effect. Generally, seasonal snowpack can be well approximated by Gaussian distribution for fully snow-covered area.

**Abstract**

Seasonal snowpack is an important predictor of available water resources in the following spring and early summer melt season. Total basin snow water equivalent (SWE) estimation usually requires a form of statistical analysis that is implicitly built upon the Gaussian framework. However, it is important to characterize the non-Gaussian properties of snow distribution for accurate large-scale SWE estimation based on remotely sensed or sparse ground-based observations. This study quantified non-Gaussianity using sample negentropy, the Kullback–Leibler divergence from Gaussian distribution, for field-observed snow depth data on the North Slope, Alaska, and three representative SWE distributions in the western US from the Airborne Snow Observatory (ASO). Snowdrifts around lakeshore cliffs and deep gullies can bring moderate non-Gaussianity in the open, lowland tundra of North Slope, Alaska, while the ASO dataset suggests that subalpine forests may effectively suppress the non-Gaussianity of snow distribution. Thus, non-Gaussianity is found in areas with partial snow cover and wind-induced snowdrifts around topographic breaks in slope and other steep terrain features. The snowpacks may be considered weakly Gaussian in coastal regions with open tundra in Alaska and alpine and subalpine terrains in the western US if the land is completely covered by snow. The wind-induced snowdrift effect can be potentially partitioned from the observed snow spatial distribution guided by its Gaussianity.

# 1 Introduction

Modeling of the spatial variability of snow is important for large-scale earth surface modeling since atmospheric circulation is sensitive to snow cover presence (e.g., Aas et al., 2016; Meng, 2017; Mott et al., 2015, 2017; Nitta et al., 2014; Younas et al., 2017). Since subgrid variability often causes appreciable bias in weather predictions, accurate snow cover quantification can potentially improve the predictability of weather, planetary boundary-layer evolution, convective cloud formation, and even tropical cyclogenesis (Santanello et al., 2018). Hence, the subgrid variability of snow cover has been incorporated into operational regional weather forecasting models such as the High-Resolution Rapid Refresh (HRRR) model (He et al., 2021).

Observations of seasonal snow storage in mountainous areas through remote sensing and ground-based measurements are a direct and reliable indicator of the water supply during the following spring season in downstream regions (e.g. Fleming et al., 2023; Sengupta et al., 2022). However, total basin snow water equivalent (SWE) estimation usually requires a statistical relationship such as the snow depletion curve (SDC), which correlates with observables such as the snow cover area fraction (SCF). Based on a study of the observed snow distributions in Reynolds Creek Experimental Watershed in Idaho, Luce et al. (1999) showed that one snow distribution can reasonably represent the SDC evolution for the rest of the season. Also, Luce and Turboton (2004) showed a high degree of similarity in nine years of dimensionless depletion curves measured in the same basin. Shamir and Georgakakos (2007) demonstrated the consistency of SDC over a season in the American River using a distributed model. The subseasonal and

interseasonal consistency in SDCs suggests the possibility for subgrid snow characterization as well as SWE estimation from SCF data such as the MODIS product (Hall et al., 2006).

As remote sensing technologies advance, seasonal snow distribution characterization becomes more approachable with multi-sensor methods. For example, Tarricone et al. (2023) analyzed three Interferometric Synthetic Aperture Radar (InSAR) image pairs to assess SWE evolution using the snow-focused multi-sensor method with Uninhabited Aerial Vehicle Synthetic Aperture Radar (UAVSAR) and an L-band InSAR data as well as optical fractional snow-covered area (SCA) information. However, to estimate the total basin SWE in water resource management practices, statistical empirical relationships are required to fill gaps in the spatial and temporal resolutions—even when using these remote sensing observations (Tsang et al., 2022). For example, Meloche et al. (2022) assumed log-normal distribution to represent the sub-pixel variability of remotely sensed data. Thus, uncertainty and subgrid variability must be accounted for when using statistical characterization in SWE estimation.

The most popular choice for the probability density function (PDF) of snow is log-normal distribution, which inherently eliminates negative snow depth (Donald et al., 1995; Liston, 2004; and many others). Brubaker and Menoes (2001) chose a beta distribution, while Kolberg and Gottschalk (2006) selected a two-parameter γ-distribution. Although these common distributions are in the exponential family, they were primarily chosen for convenience. Indeed, the representativeness of these parametric probability distributions remains questionable for different landscapes and snowpack ages (e.g., Skaugen & Randen, 2013; Egli & Jonas, 2009; He, Ohara, & Miller, 2019). Moreover, these approaches for bounded distributions may not work for evolving snowpacks with partial SCA where zero values are present in the probability domain.

In theory, since the landing location of each snow particle fallen from the atmosphere is considered an independent and identically distributed (iid) random variable, the resulting snow depth or SWE distribution should asymptotically approach a Gaussian distribution due to the central limit theorem. He, Ohara, and Miller (2019) reported Gaussian snow distributions in many of the forested, fully snow-covered areas during the peak snow season using airborne Light Detection and Ranging (LiDAR) observations in the Snowy Range, Wyoming. This implies the presence of both systematic (non-Gaussian) and random (Gaussian) mechanisms in snow accumulation and ablation processes. Therefore, it is possible to identify the potential factors as “signals” that make the snow distribution deviate from a Gaussian distribution by analyzing the resultant snow distributions.

This study applies negentropy to analyze the non-Gaussianity of snow distributions in Arctic tundra, as well as alpine and subalpine landscapes in North America. Negentropy measures the departure in entropy between a sampled distribution and Gaussian distribution of identical variance and mean. Signals of interest (e.g., systematic snowdrift patterns) can be extracted as non-Gaussian components because pure random noise asymptotically becomes Gaussian in theory. This is the main idea of independent component analysis (ICA; Hyvärinen et al., 2000). This work presents the quantified non-Gaussianity of the observed snow distributions through a variety of snow distribution data, including intense direct hand measurements within 30-m grids using a probe, and indirect measurements using a snowmachine-attached ground-penetrating

radar (GPR), UAV-based photogrammetry, as well as the Airborne Snow Observatory (ASO) SWE products.

# 2 Methods

## 2.1 Negentropy

To measure the non-Gaussianity of any data, we implement the information-theoretic metric of negentropy as the objective function since negentropy is equal to the Kullback–Leibler divergence between $p_x$ and a Gaussian distribution with the same mean and variance as $p_x$. There is a well-known proposition that Gaussian density has the largest information entropy among all unbounded distributions with the same first and second-order statistics. As such, the non-Gaussianity of an observed distribution can be quantified by negentropy *J*, which is defined as follows (Hyvärinen et al., 2000):

$$J(X) = S\left(X_{gauss}\right) - S(X) \tag{1}$$

where S is the information entropy of *X*. The information entropy can assume a diversity of metrics ranging from the most general capturing microphysical event-scale codependence in nonlinear statistical mechanics (Perdigão 2018) or simply assuming basic event-scale independence in classical information theory (Shannon (1948) statistical entropy). For the purpose of this study, we take the latter simple form, which is defined as:

$$S(X) = -\int p_x(\eta) \log[p_x(\eta)] d\eta. \tag{2}$$

The Edgeworth expansion (Edgeworth, 1905) can provide an approximation for a PDF of X, as follows:

$$p_x(X) = \frac{\phi(U)}{\sigma}\left[1 + \frac{\kappa_3}{6} H_3(U) + \frac{\kappa_4}{24} H_4(U) + \frac{{\kappa_3}^2}{72} H_6(U) + \cdots\right] \tag{3}$$

where

$U$ = standardized random variable of $X$

$H_k(U)$ = Chebyshev-Hermite polynomials

$\phi(U)$ = standard normal density

$\kappa_k$ = k-th order cumulant of $U$.

Substituting the Edgeworth series into the negentropy definition, Comon (1994) obtained the analytical expression:

$$J(X) = \frac{1}{12}{\kappa_3}^2 + \frac{1}{48}{\kappa_4}^2 + \frac{7}{48}{\kappa_4}^4 - \frac{1}{8}{\kappa_3}^2\kappa_4 + O(n^{-2}). \tag{4}$$

This is the estimator of negentropy at fourth-order cumulant. In practice, a more intuitive approximation is commonly used, as follows:

$$J(X) = \frac{1}{12}\text{skew}(U)^2 + \frac{1}{48}\text{kurt}(U)^2 \tag{5}$$

where skew and kurt are the skewness and kurtosis of standardized variable, *U*, respectively.

The sample estimation of the higher-order moment and cumulant (e.g., skew and kurtosis coefficients) is known to be sensitive to the presence of outliers. In this study, the interquartile range (IQR) method was adopted for outlier removal with a minimum removal that lies outside the range of three times IQR.

While negentropy metrics and corresponding Edgeworth approximations have been previously explored and further developed in atmospheric sciences and in physics, including derivations and implementations to higher-order distributions, elaborate numerical and analytical estimators (Pires and Perdigão 2007, Perdigão 2010, Perdigão 2017), the present study brings a simplified treatment not yet explored in Hydrology and tailored for swift and seamless integration within hydrological and water resource systems investigations.

### 2.2 Data collection

We analyzed four types of data with different collection methodologies at various scales in this study. The first is manual snow depth surveys using a GPS-aided snow probe (Magnaprobe; Sturm & Holmgren, 2018), the second is snow depth transects using a snowmachine-attached GPR, the third is snow depth maps using UAV-based photogrammetry, and the last is the SWE product of the ASO. The first three datasets are for the open tundra in the Arctic Coastal Plain (ACP) of Alaska while the ASO data are for the alpine and subalpine regions of the continental USA. Detailed data specifications associated with the collection methodologies will be presented in Results section below. Figure 1 displays the map of the snow depth surveys in North Slope, Alaska, USA.

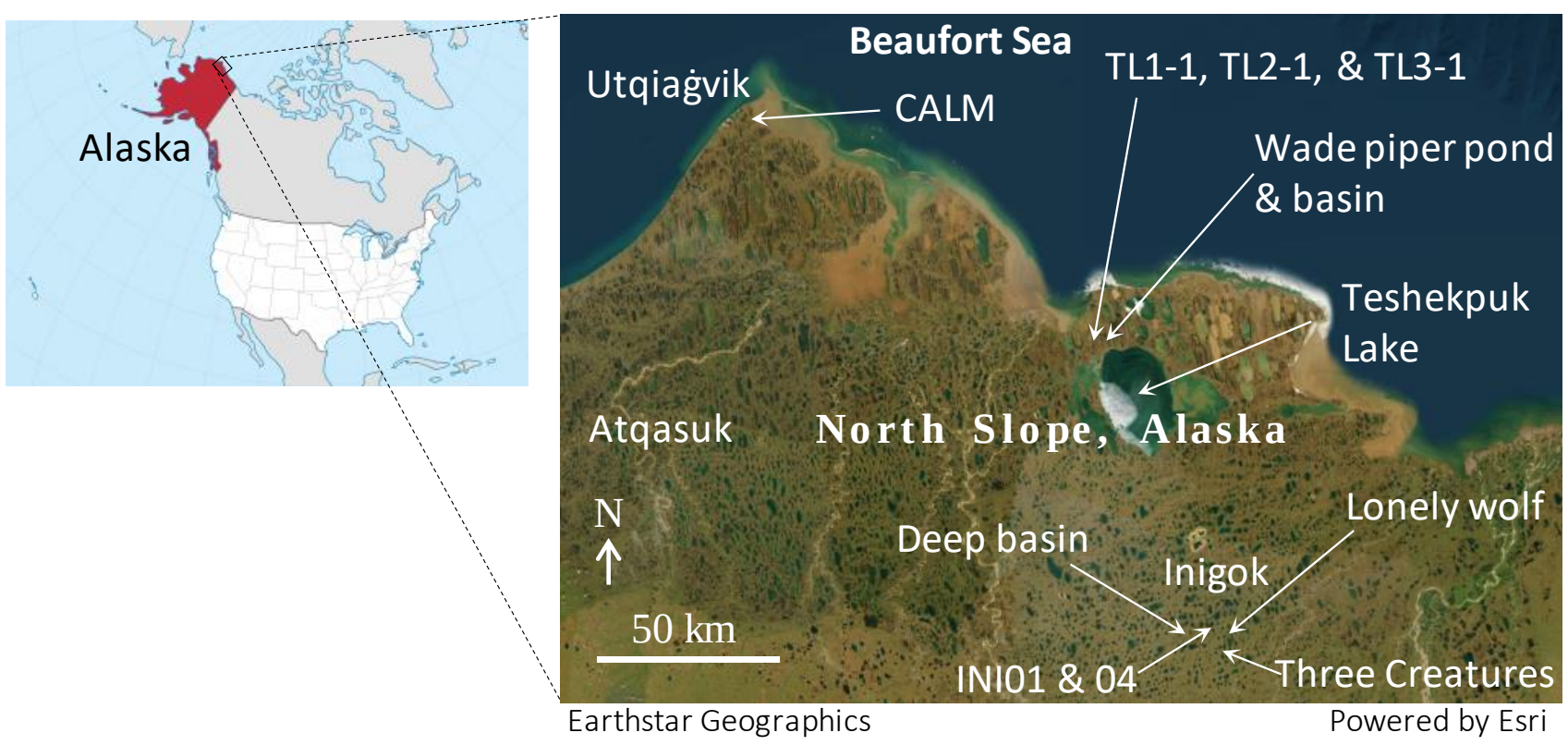

Figure 1: Map of the snow survey locations in Alaska, USA.

# 3 Results

### 3.1 Manual snow surveys at Teshekpuk, North Slope, Alaska (May 2022)

Snow depth data were collected using a Magnaprobe (Sturm & Holmgren, 2018) in five 30 x 30-m grids with 1 m grid spacing north of Teshekpuk Lake, North Slope, AK, in May 2022. The GPS location of each measurement was automatically recorded. Figure 2 presents the

interpolated snow depth distributions and corresponding histograms (right columns) in five areas near Teshekpuk. The observer measured the point scale snow depth at approximately every 1 m along a line toward flags placed 1 m apart on the surface. Since the data points were selected from undisturbed snow, the locations are unevenly distributed despite the snowpacks generally being highly hardened by wind. The relative spatial locations are considered accurate since the operator stood on the same side of the probe and followed pre-marked lines based on the tape measure; however, the absolute plotted coordinate in the figures may not be trustworthy due to the GPS horizontal accuracy < 3 m.

The graphics in the left column of Figure 2 present the point depth observation locations and interpolated snow depth distributions using the nearest distance method. The number of data points denoted by the black dots is n=951 (TL1-1), n=925 (TL2-1), n=904 (TL3-1), n=927 (Wadepiper Pond), and n= 960 (Wadepiper Basin).

The corresponding snow depth histograms and three fitted distributions are displayed in the right column. The statistics mean, standard deviation, skew coefficient, and negentropy ($J$) are reported on the top part of each graph. In general, the snow depth distributions in these areas are almost Gaussian distributions since the computed negentropy is small. However, the negentropy of snow distribution affected by wind-induced snowdrift (sastrugi) on frozen lakes is larger than the tundra covered by sedge and herbaceous vegetation. In practice, the non-Gaussianity of seasonal snow depth may have been negligible in the coastal open tundra (including frozen open waters) in the Teshekpuk study area in May 2022.

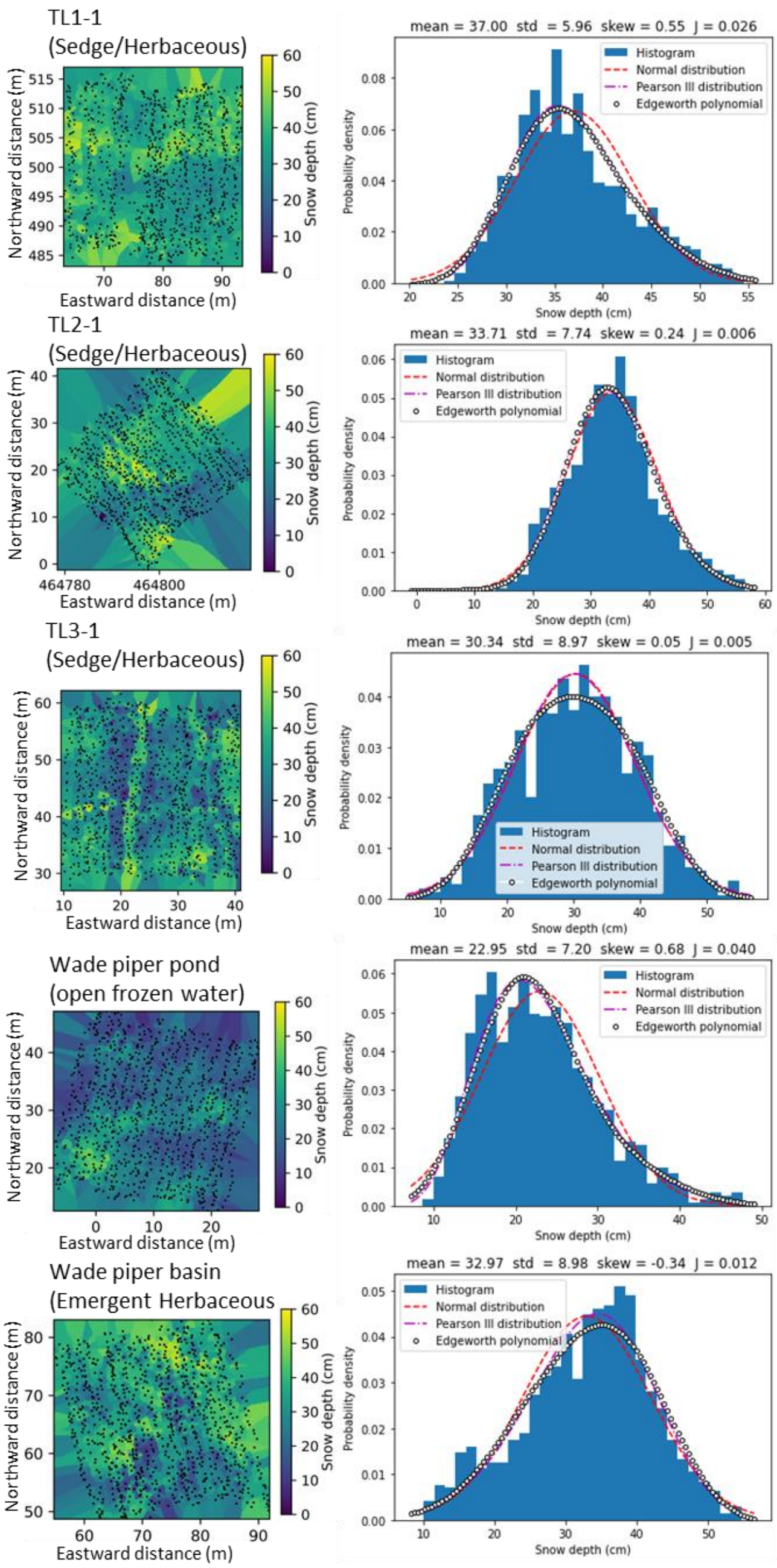

Figure 2: Manual snow distributions in the Teshekpuk Lake area, North Slope, Alaska (May 2022) and corresponding histograms with fitted probability density functions (PDFs). *J* denotes the computed negentropy. Snow depth histograms in open tundra in 30 m x 30 m squares are generally categorized as "weak-non-Gaussian." The approximated center coordinates of the grids

are 70.738°N, 153.970°W (TL1-1), 70.740°N, 153.956°W (TL2-1), 70.739°N, 153.928°W (TL3-1), 70.751°N, 153.870°W (Wadepiper Pond), and 70.746°N, 153.854° W (Wadepiper Basin).

### 3.2 Snow depth surveys using GPR along multiple transects in Inigok, North Slope, AK (April 2019)

The Inigok area of North Slope, Alaska (70.001° N, 153.068° W) is characterized by paleo sand dunes (Carter, 1981), hydro-geomorphological processes, and permafrost landforms such as thermokarst lake formation and drainage. The landscape is characterized by relatively steep terrain and substantial wind-induced snowdrifts (deeper than 5 m), especially around lake shores and drainage channels (e.g., Rangel et al., 2023).

Snow depth surveys using a GPR are particularly effective for deep-snow areas since the Magnaprobe is only 1.5 m long. Considering the lower limit of the selected GPR antenna, we collected several GPR transects (Malå ProEx, 800 MHz, GuidelineGeo, Sundbyberg, Sweden) around Inigok, where the snowpack was deeper than in the coastal area. The antenna was placed on a sled towed by a snowmachine traveling < 5km $h^{-1}$. The effect of compaction by the snowmachine was considered negligible because the snow was highly wind-packed and therefore was not affected by the weight of the snowmachine during data collection. The GPR data were processed in ReflexW (Sandmeier Software, Karlsruhe, Germany) using a low frequency noise removal (dewow) and a linear gain with topographic correction adapted from the ArcticDEM (Rangel et al., 2023b). Maps of snow depth estimated from the GPR transects are shown in Figure 3. The line color denotes the observed snow depth (the darker, the deeper). A substantial snowdrift developed near the lakeshore's banks due to its steep topography.

Figure 4 displays the histograms of GPR snow depth data in Inigok, North Slope, Alaska, in May 2019 when using (A) all transect data and (B) the frozen lake sections only. The snow depth histogram of all transects shows strong non-Gaussianity due to a mix of steep and flat terrain. However, the histogram of the partial dataset only for the frozen lakes shows much weaker non-Gaussianity. In fact, snow distribution after removing the deep-snow parts can be reasonably approximated by the Gaussian distribution with a negentropy of 0.037, which is the same level as Wadepiper Pond (Figure 2) in the previous section (J = 0.040). Therefore, the snowdrift due to steep terrain is considered a major source of non-Gaussianity in snow depth in open tundra.

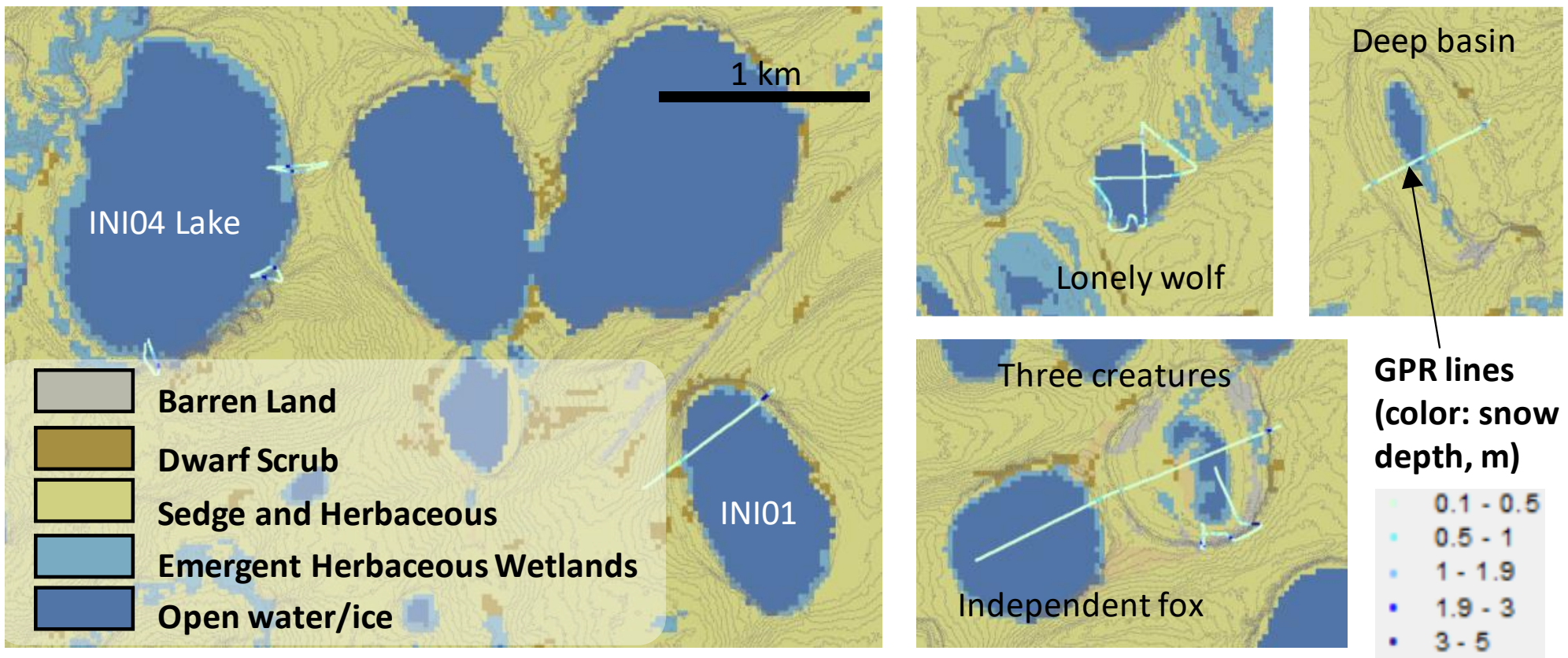

Figure 3: Snow depth surveys using GPR along multiple transects in Inigok, North Slope, Alaska (27 and 28 April 2019). The approximated center coordinates of the maps are 153.105W 70.005N (INI04 & INI01), 152.949W 69.993N (Lonely wolf), 153.274W 69.992N (Deep basin), and 153.032W 69.942N (Three creatures & Independent fox).

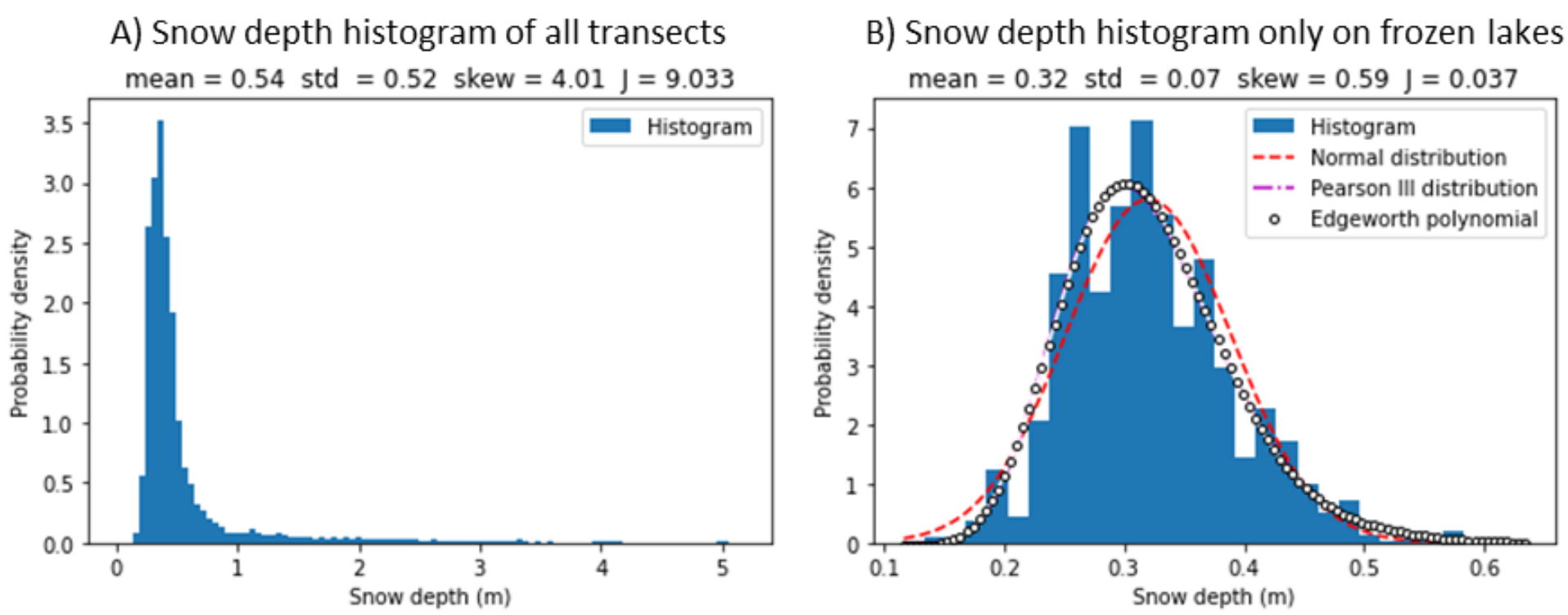

Figure 4: Snow depth histograms of GPR snow survey data from Inigok, North Slope, Alaska (April 2019) using A) all transects and B) sections on frozen lake only. Snow distributions in the Inigok area are highly non-Gaussian, while the frozen lake subset shows weak non-Gaussianity.

### 3.3 Snow depth distribution based on UAV footage of a drained lake basin within the CALM 1-km grid near Utqiaġvik, AK (May 2019)

Figure 5 (left panel) presents the observed snow distribution of a drained thermokarst lake basin referred to as Central Marsh, part of the Circumpolar Active Layer Monitoring (CALM) Network located east of Utqiaġvik, Alaska. The snow depth was estimated by differentiating the snow surface elevation and the snow-free ground elevation using UAV surveys with the photogrammetry technique. The images were collected on August 4, 2019 (snow-free), and April 15, 2019 (snow-covered), using a Phantom 4 UAV (P4RTK). Images were post-processed/georeferenced to NAD83 Zone 4 North in ellipsoid heights using a propeller aeropoint

and Pix4D (version 4.3.33 for the April survey, 4.4.12 for the August survey) at 0.25 m spatial resolution (Nichols, 2020). The vertical accuracies of these measurements are 18 cm and 10 cm for the April and August surveys, respectively. The horizontal resolution for the snow depth is 1 m.

The CALM site is situated in the ACP in northern Alaska, which has typical complex terrain due to the recently drained thermokarst lake with sparse or negligible vegetation and well-developed polygons. There is an obvious smoothed bluff in the center of the domain, and the west side of this bluff tapers into the drained lake basin. The incised drainage channels cause steep land features that capture sizable snowdrifts in the southern part. In the southern portion of the area, the polygons are formed by ground surface cracks with ice wedge development beneath.

The negentropy distribution in the moving window may be obtained from this gridded snow data at a very high spatial resolution. The right panel of Figure 5 presents the computed negentropy map in the CALM area with a 30-m moving window. Overall, non-Gaussianity in the CALM site was found to be weak—even with the smoothed bluff and despite high snow depth. However, as whiter parts in right panel of Figure 5 are found along the drainage channels, topographic discontinuity around the incised gully seems to cause significant non-Gaussianity. Additionally, vegetation patches may bring spotty non-Gaussianity in the northern part of the area. Conversely, since the southern parts covered by the polygons except the drainage channels show darker color ($J < 0.025$), the ground surface polygon does not make snow distribution non-Gaussian. Overall, snowpack in the coastal parts of the ACP can reasonably be approximated by Gaussian distribution since most of the CALM area showed a small negentropy of less than 0.2.

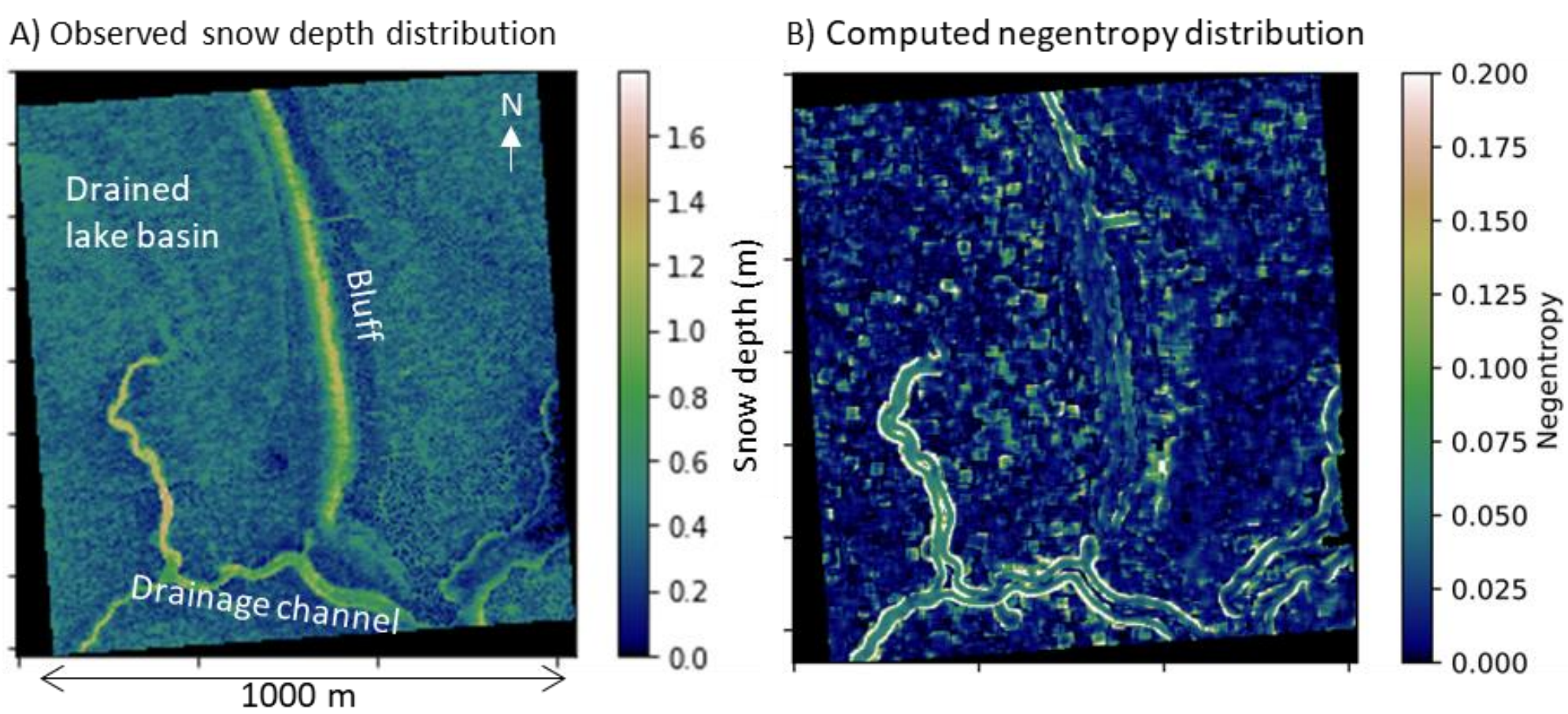

Figure 5: Snow depth distribution based on UAV photogrammetry and the computed negentropy distribution of 30-m moving windows in a drained lake basin in the CALM 1-km grid (71.3026° N, 156.6008°W) near Utqiaġvik, Alaska.

Figure 6 presents the snow depth histogram, which looks like a Gaussian distribution with a long tail due to snowdrift around the gullies in the CALM grid. In fact, when the deep snowdrifts of

the gully and the bluff are removed from the samples, the histogram more closely resembles a Gaussian distribution (see the right panel in Figure 6).

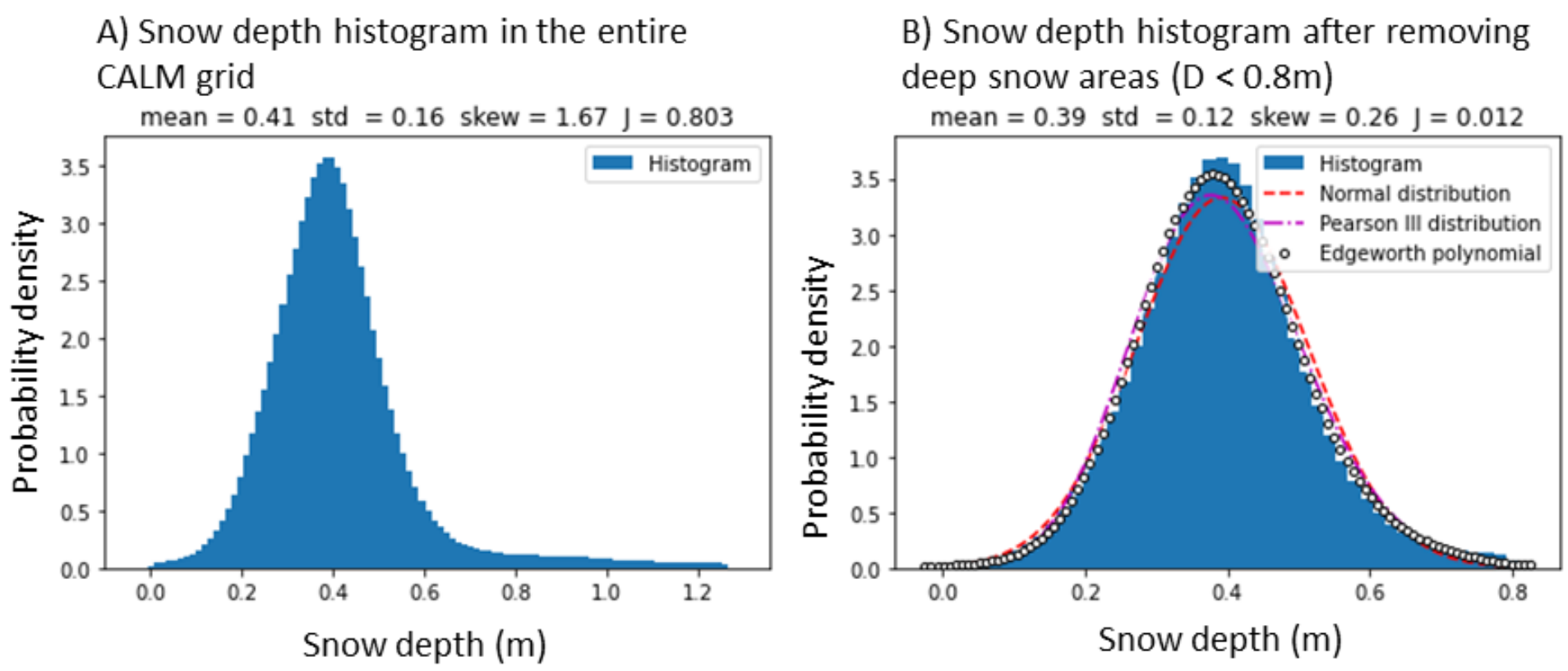

Figure 6: Snow depth histogram based on the UAV photogrammetry of a drained lake basin in the CALM 1-km grid near Utqiaġvik, Alaska. Removing the deep snow parts caused by wind-induced snowdrift results in a near-perfect fit by Gaussian distribution.

### 3.4 SWE products based on ASO data for the selected watersheds

SWE is a stable and direct indicator of snow/water distribution in landscapes. As such, the SWE products from the Airborne Snow Observatory (ASO) were selected (Painter et al., 2016) to examine the Gaussianity of snow distributions in different climate zones and landscapes with alpine to subalpine snowpack. The snow depth and SWE distributions were estimated from the coupled imaging spectrometer and scanning LiDAR, then combined with distributed snow modeling (including snow density simulation). The ASO snow products are considered one of the most comprehensive instantaneous snow distribution estimations at fine resolution (50 m). We used the processed snow product to characterize the medium-scale snow distribution with the same outlier treatment (IQR method) as described above.

The analysis of three representative SWE datasets in the western US is presented. These include Upper Tuolumne River watershed in California (USCATB, April 3, 2013), East River watershed above Gunnison, Colorado (USCOGE, March 31, 2018), and the Olympic Mountains in Washington (USWAOL, March 29, 2016).

#### 3.4.1 Tuolumne River Watershed, California

Figure 7 presents the composite graphics of the data and the analysis results for the Upper Tuolumne River watershed on April 3, 2013. Panel A shows the SWE distribution estimated by the ASO, while panel B visualizes the normalized SWE histogram or PDF within the entire domain with the fitted theoretical distributions. Panels C and D are the negentropy distributions of the SWE within 1500-m moving windows with and without partially snow-covered cells. Panel E shows the NLCD 2011 land cover map for reference. The watershed (1175 km$^2$) is one of the drainages to the California Central Valley through the Hetch Hetchy reservoir in the southern Sierra Nevada Mountain Range. The boundary of the catchment is mostly comprised of

steep rocky alpine terrain (which contributes to the attractive land features of Yosemite National Park), whereas the bottom of the valley is relatively flat due past glacial processes. The snow distribution (panel A) shows a clear relationship with elevation, while the SWE barely exceeded 1 m during the observation period in peak snow season. The overall SWE histogram (panel B) illustrates strong non-Gaussianity because of snow-free and shallow accumulation areas in the watershed (bounded distribution effect).

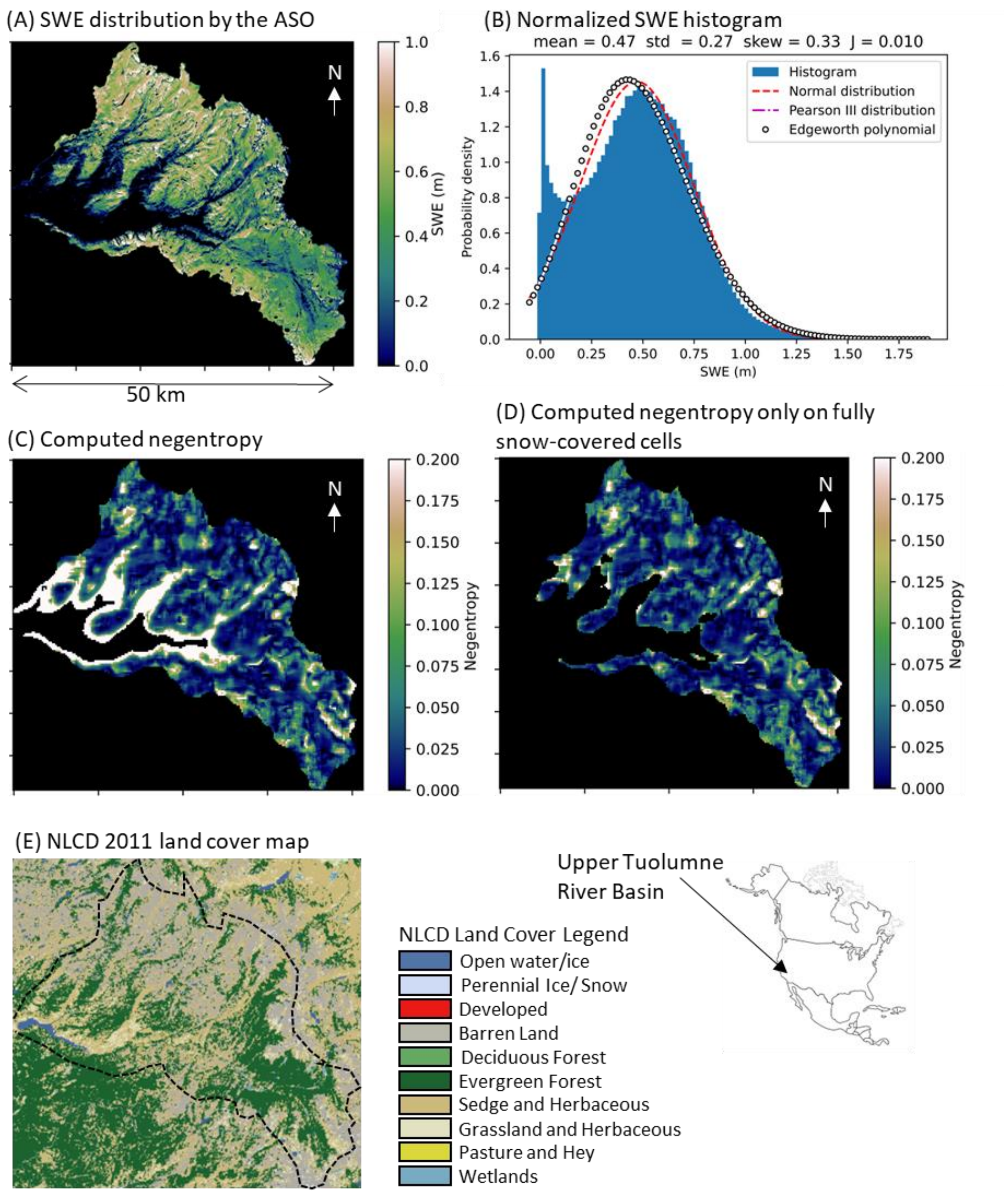

Figure 7: (A) SWE distribution based on ASO data of the Upper Tuolumne River Basin, California, USA from April 3, 2013 (USCATB, 37.461°N, 119.494°W); (B) normalized SWE histogram; (C) negentropy map of the SWE within 1500-m moving windows; (D) negentropy map of only fully snow-covered cells; (E) NLCD 2011 land cover map.

However, the local negentropy map with moving windows (panel C) shows small non-Gaussianity except in the low-elevation areas. In fact, the majority of high non-Gaussianity cells are from partially snow-covered cells. When the partially snow-covered cells are removed in panel D, the local negentropy falls by less than 0.15 in most of the watershed. Therefore, the bounded distribution effect in the probability domain from the partially snow-covered cells brings substantial non-Gaussianity into the snow distribution.

### 3.4.2 East River, Colorado

The ASO dataset of the East River above Gunnison, Colorado (USCOGE) was selected as a representative basin in the Rocky Mountains region. This dataset includes the U.S. Department of Energy (DOE)'s East River community observatory, where comprehensive field data have recently been collected (Kakalia et al., 2020). The data domain, which does not agree with the watershed boundary, is approximately 1670 km$^2$ with the elevation ranging from 2,343m (Gunnison) to 3,901 m.  Figure 8 displays the corresponding analysis results of the East River area on March 31, 2018.

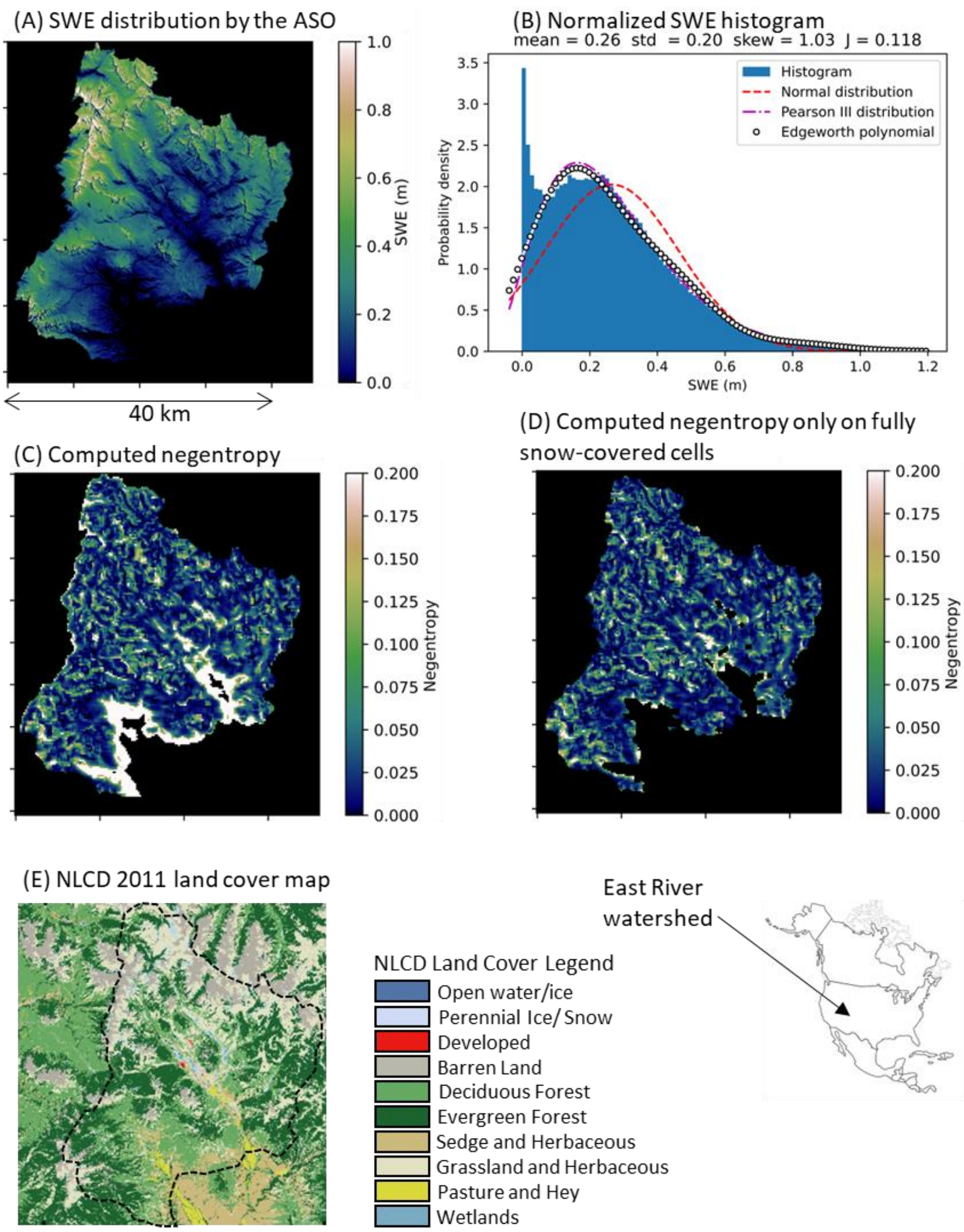

Figure 8: (A) SWE distribution based on ASO data for the East River watershed above Gunnison, Colorado, USA from March 31, 2018 (USCOGE, 39.037°N 106.978°W); (B) normalized SWE histogram; (C) negentropy map of the SWE within 1500-m moving windows; (D) negentropy map of only fully snow-covered cells; (E) NLCD 2011 land cover map.

Besides the obvious bounded distribution effect of partially snow-covered cells, this case study illustrates the non-Gaussianity induced by the steep topographic features around the high peaks in the Rocky Mountains. Also, since the lower negentropy (darker colored) parts in panel D

generally agree with the evergreen and deciduous forest cover extent in the NLCD land cover map in panel E, the subalpine forest may reduce non-Gaussianity in snow distribution. However, the general characteristics of the sample's negentropy distribution in Upper Colorado are consistent with the Upper Tuolumne River watershed in the Sierra Nevada Mountain Range.

### 3.4.3 Olympic Mountains, Washington

The last example of snow non-Gaussianity quantification is the Olympic Mountains in Washington, USA, which represent the Northern Pacific Coastal Range under strong oceanic influence. The elevation ranges from sea level to 2430 m. The Olympic Mountains consist of a cluster of steep-sided peaks, heavily forested foothills, and incised deep valleys. The ASO data have a large spatial coverage (5,330 $km^2$) when compared to the other two ASO datasets presented here.

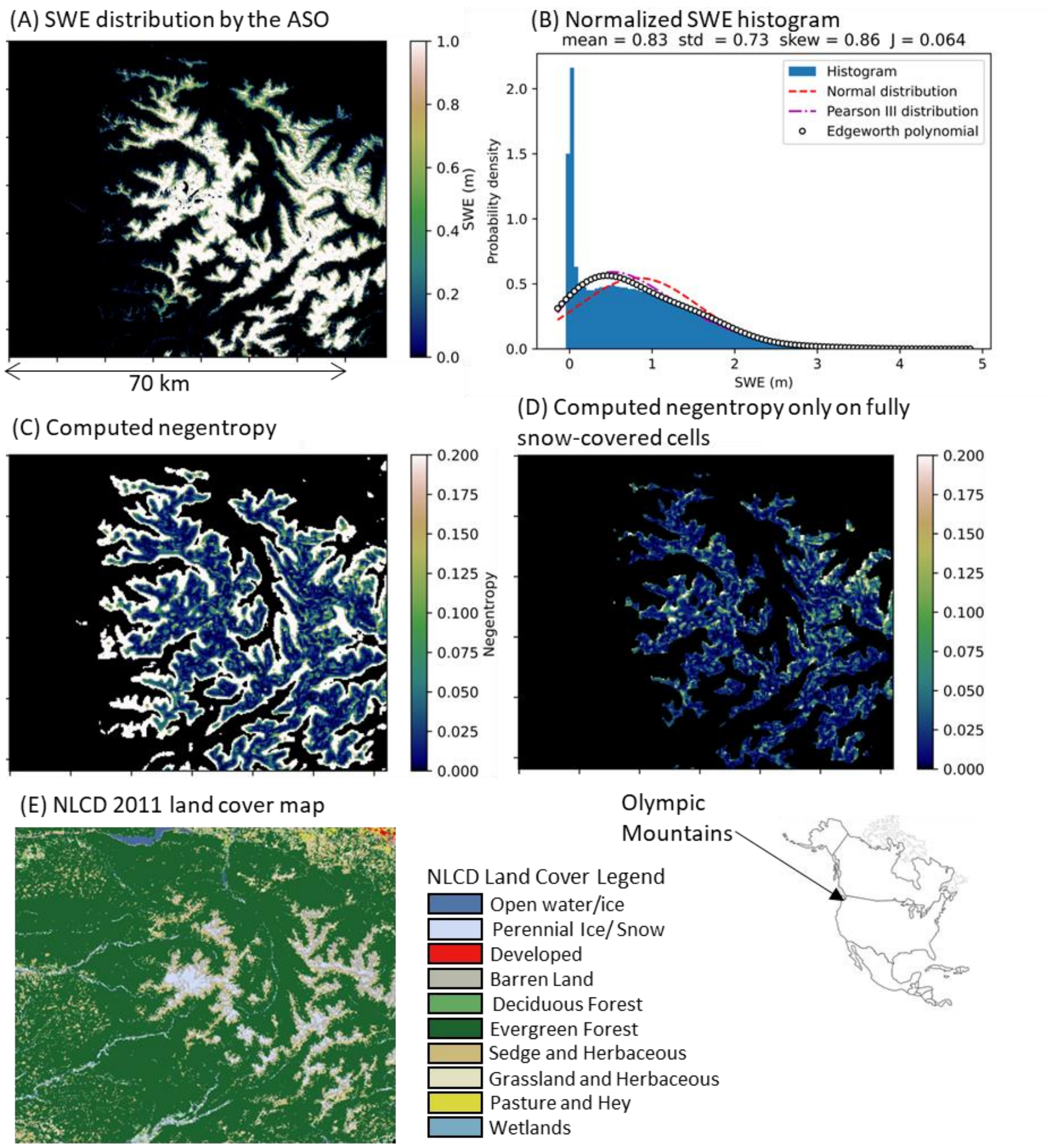

Figure 9: (A) SWE distribution based on ASO data for the Olympic Mountains, Washington, USA from March 29, 2016 (USWAOL, 47.792°N 123.650°W); (B) normalized SWE histogram;

(C) negentropy map of the SWE within 1500-m moving windows; (D) negentropy map of only fully snow-covered cells; (E) NLCD 2011 land cover map.

The black areas in the high elevation range in panel A are the approximate glacier extent excluded from the analysis (Painter et al., 2018). A large fraction of partially snow-covered cells also introduces non-Gaussianity in SWE in this region. Meanwhile, dense evergreen forests in the Olympic Mountains seem to effectively reduce the non-Gaussianity of SWE above the snow line during the ASO scanning period. Overall the non-Gaussianity of the snowpack may be considered small when compared to the other two examples, which is likely due to denser forest cover. Presumably, the vegetation cover minimizes the wind-induced snow redistribution process and makes the snow distribution more Gaussian. These characteristics—i.e., non-Gaussianity in partially snow-covered areas and high Gaussianity in forested areas—are common features of the SWE distributions throughout the western US.

# 4 Discussion

The sample negentropy values presented here are generally consistent with each other despite the variety of data collection methods used at different scales. The level of random noise in the datasets depends on the data collection methods. Among the datasets discussed here, one may anticipate that the ASO data have the largest Gaussian bias due to multiple remote sensing, resampling, assimilating, and modeling procedures covering remarkable spatial coverages with uniform data quality. The UAV-based LiDAR data at the North Slope CALM site are expected to have a noticeable random bias with a vertical accuracy of approximately 12 cm. The GPR snow depth observations should have a smaller but appreciable Gaussian bias due to snow quality variation and non-flat snow surface elevation (antenna angle vibration), although the continuous measurement minimizes the random relative error in the snow depth estimation. The hand-measured snow depth data using a probe may include the least Gaussian bias, while the sampling spacing was not uniform and in addition, due to relative poor spatial positioning control with the Magnaprobe's onboard GPS unit. Despite these differences, it is encouraging that the quantified Gaussian levels were comparable and consistent since they share common features.

The stability of the sample estimator of negentropy may be a potential issue, especially when the sample size is small. Additionally, since the higher-order statistical moments are sensitive to the presence of outliers in the sample, an outlier removal filter is recommended for large samples. The IQR method with a threshold of 3 IQR above the third quarter (Q3), which is much stricter than the usual threshold (typically 1.5 IQR), has been applied for the UAV photogrammetry data and the ASO datasets for computational stability. Even with the large threshold (small outlier removal), the proposed method using negentropy appears to be effective in characterizing the Gaussianity of snow distribution, which has been a common implicit assumption for existing gridded data and models. This study visualized the limitation of such a common distribution assumption for snow distribution, specifically for areas with partial snow cover.

To summarize the analyses presented here, five categories of Gaussianity were defined and associated with a magnitude of sample negentropy value (see Table 1). Most of the fully snow-

covered areas fell into the category "almost Gaussian," with a negentropy less than 0.03. Notably, a negentropy less than 0.01 is considered nearly perfect Gaussian, as can be seen in the previous sections.

The Gaussianity characterization of snow distribution appears to be useful in distinguishing the snowdrift-affected areas using the sample negentropy. Simultaneously, this finding can justify the implicit Gaussian assumption for snow distribution for overall SWE estimation, particularly for snowpack characterization from remotely sensed information. For instance, the effect of higher-order statistical moments can be negligible in most fully snow-covered areas. Conversely, some additional statistical treatment for higher order statistics may be required for the areas with the non-Gaussian effects around snow lines, open wind-swept areas, and sharp terrains. Additionally, since consistent pattern in skew coefficient was not identified from the snow datasets, the commonly-used log-normal distribution may not be suitable for those areas.

Table 1: Summary of the analysis using the sample negentropy.

| **Class** | **Negentropy** | **Landscape & land cover type** | **Examples** |
|---|---|---|---|
| Strong non-Gaussian | $0.2 < J$ | Partially snow-covered areas, mixture of landscapes (steep-flat) | CALM, Inigok, Upper Tuolumne, East River, Olympic Mountains |
| Non-Gaussian | $0.1 < J \leq 0.2$ | Snowdrift around steep terrain | CALM |
| Weak non-Gaussian | $0.03 < J \leq 0.1$ | Snowdrift on a frozen lake, vegetation cluster | Teshekpuk, Inigok, CALM |
| Nearly Gaussian | $0.01 < J \leq 0.03$ | Most of the uniform terrain in open tundra and alpine forest | Teshekpuk, CALM, Upper Tuolumne, East River, Olympic Mountains |
| Gaussian | $J \leq 0.01$ | Open tundra (sedge, polygons), most forested areas | Teshekpuk, Upper Tuolumne, East River, Olympic |

## 5 Conclusions

A Gaussian snow distribution is a common underlying assumption for finite scale models or gridded datasets. The present study tested this assumption using the sample negentropy of various snow data. We found two main sources of non-Gaussianity: (1) partial snow cover effect (bounded distribution) and (2) wind-induced snowdrift effect around steep terrain features. The second effect may amplify the first one in wind-swept alpine areas since snow erosion remains shallow on rocky ridges and peaks. The snowdrift around lakeshore cliffs and deep gullies can bring moderate non-Gaussianity in the open tundra of North Slope, Alaska. However, the wind-

packed snow in the coastal plain region of the North Slope may generally be categorized as weakly Gaussian during mid to late winter due to the continuous snow cover. This implies that the non-Gaussianity of the snowpack may not be neglected during the snow accumulation season and late spring season. Interestingly, small ground surface features (e.g., low-centered and high-centered ice wedge polygons) make snow distribution more Gaussian, while snowdrift (snow dunes) on a flat frozen lake seems to be less Gaussian than on tundra or in a drained lake basin.

Our analyses of the ASO SWE products reinforced the findings for snowpacks on the tundra. Although SWE data was chosen instead of snow depth for practical reasons, the common features in non-Gaussianity remain valid. Additionally, the snow diffuser effect of forests was illustrated in three representative areas in the western US. This effect was reported by He et al. (2019) based on airborne LiDAR snow depth measurements on the Snowy Range, Wyoming, USA. Hence, it is likely that vegetation cover generally makes snow distribution more Gaussian in the snow accumulation process; however, further verification of this relationship is recommended.

Overall, a Gaussian distribution is a suitable approximation for snow spatial distribution when the ground is completely covered by snow. Higher-order statistics associated with landscape type may potentially improve the SWE estimation in wind-swept open terrain and near snow lines. The level of non-Gaussianity will determine the choice of statistical tool to correct the systematic bias in snow measurements. Meanwhile, this study suggests the possibility of partitioning the extent of wind-induced snowdrifts by means of independent component analysis (Comon et al., 2010).

# Author contribution

NO performed the analysis, and RAPP offered technical advice. NO, ADP, RCR, and BMJ provided the field observed data for the case studies in Alaska. ADP, BMJ, KMH, RAPP, and RCR actively participated in the discussions and manuscript improvement. NO prepared the manuscript with contributions from all co-authors.

# Competing interests

The authors declare that they have no conflict of interest.

# Acknowledgments

This study was supported by the National Science Foundation (NSF) Office of Polar Programs (OPP) under the awards 1806287, 1806213, and 1806202. The authors thank UIC Science and CH2MHill Polar Field Services (now Battelle Arctic Research Operations) for their logistical field support. RP acknowledges support from the European Union under the Horizon Europe grant 101074004 (C2IMPRESS), the Meteoceanics flagships MR-220617 and MR-070220-BLUE. We used the NSIDC DAAC Airborne Snow Observatory (ASO) data downloaded from NSIDC.org.

# Data Availability Statement

The data used in this research are publicly available at the NSF Arctic Data Center: https://doi.org/10.18739/A24746T0K, and https://doi.org/10.18739/A2NV99C4P

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
