# Peer review of "Characterization of Non-Gaussianity in the Snow"

_EGUsphere, 2024_

## Referee Comment (RC1)

**Review of preprint manuscript** : https://doi.org/10.5194/egusphere-2024-395 by Ohara et al.
**June 2024.**

**General comment**

I thank the authors for the nice and easy-to-read manuscript. In my opinion the research topic is of high interest at the area of blooming remote-sensing products for snow. The data and methods are well described (except a few minor missing informations that I detailed below) and the analyses are sound.

Besides some minor formal suggestions that you will find below, my main comments are related to the implications of the research carried out, that could imo be better described or enlarged, for the benefit of the impact of the paper and appropriation of its findings by a wider snow research community. The implications for the subgrid parameterizations of snow depth / SWE for snow or hydrological models are stated but could be described in more details (what are the current assumptions prevailing in models for this, are there different below forest vs in open areas, how do the paper's findings impact on them ?) In general the described applications of the paper's finding should be described more in-depth. I think there could be also implications related to the assimilation of station snow depth data within operational hydrological models. This remark pertains both to the Introduction and to the Discussion or conclusion parts.

Overall my appreciation of the paper is positive and I encourage its publication provided the above main comment and the following minor comments are addressed.

**Minor comments**

P2 L44 : you could cite here a bit more literature in support of this statement and extend it to regional climate modelling (for instance citing *Rudisill et al 2024, Lalande et al 2023*)

P2 L56 : Luce and Turbonton → Tarboton

P3 L 61 : SWE → bassin-wide SWE

P3 L83 : I am not a statistician expert, but I would argue that the landing location is affected by micro-topography and meteorological effects at the micro-scale (e.g preferential deposition downwind of a crest); is this compatible with "identically distributed"; isn't there a scale effect or spatial aspect to consider ?

P3 L 87-89 : " This implies the presence of both systematic (non-Gaussian) and random (Gaussian) mechanisms in snow accumulation and ablation processes.". I don't see the implication link with the previous sentences. Or rather : I see it, but I think that the meaning of "in theory" L82 should be clarified to make this paragraph clearer (If I understand correctly, all the micro-scale/topographic effects of my previous comment are excluded from the initial "in theory" of the paragraph, but this should be explicitly stated)

P5 L 142 : Hydrology → hydrology

P5 L 148 : I understand the interest of having statistics of SWE instead of snow depth for hydrological purposes, but are there other motivations behind the use of SWE instead of Snow depth from ASO data ?

P5 Fig 1 : the equivalent of Fig 1 for the non-Arctic sites would be great, as well as a table with a short description of the different sites (or sub-sites)'characteristics : extent of the data collection zone, spatial resolution of the data, estimated accuracy, date, collection method (GPR, magnaprobe, etc…), vegetation cover/variability, landform(s).

P6 and further : Some study sites lack a detailed description of topography. For instance in subsection 3.1 too little info is given on this aspect ; Fig 3 entails iso-altitude lines but we don't know their altitude spacing ; line 241 the polygons are mentioned but we learn only at the very end of the manuscript that there are both low-centered and high-centered.

P8 Sect 3.2 : the spatial resolution of the GPR data should be specified for comparison with other monitoring methods

Sect 3.4 : In general in this section, the effect of forest vegetation on the Gaussianity could be better highlighted by providing explicitly SWE distributions on forest-covered areas vs on other areas.

Also in this section 3.4 and further in the discussion and conclusion, the **effect of scales** should be more emphasized : snow depth/SWE on the forest floor may be quite Gaussian when looked at at the spatial scale of over a few meters, but at decimetric or centimetric scales this is likely not true.

L 370 : much stricter → less stricter seems more accurate to me (?)

L377 : I very much like this way of synthesizing your findings.

**References**

Lalande, M., Ménégoz, M., Krinner, G., Ottlé, C., & Cheruy, F. (2023). Improving climate model skill over High Mountain Asia by adapting snow cover parameterization to complex-topography areas. *The Cryosphere*, *17*(12), 5095-5130.

Rudisill, W., Rhoades, A., Xu, Z., & Feldman, D. R. (2024). Are atmospheric models too cold in the mountains? The state of science and insights from the SAIL field campaign. *Bulletin of the American Meteorological Society*.

---

## Author Comment (AC1)

Our responses are denoted in blue color below.

**Review of preprint manuscript** : https://doi.org/10.5194/egusphere-2024-395 by Ohara et al. **June 2024.**

**General comment**

I thank the authors for the nice and easy-to-read manuscript. In my opinion the research topic is of high interest at the area of blooming remote-sensing products for snow. The data and methods are well described (except a few minor missing informations that I detailed below) and the analyses are sound.

Besides some minor formal suggestions that you will find below, my main comments are related to the implications of the research carried out, that could imo be better described or enlarged, for the benefit of the impact of the paper and appropriation of its findings by a wider snow research community. The implications for the subgrid parameterizations of snow depth / SWE for snow or hydrological models are stated but could be described in more details (what are the current assumptions prevailing in models for this, are there different below forest vs in open areas, how do the paper's findings impact on them ?) In general the described applications of the paper's finding should be described more in-depth. I think there could be also implications related to the assimilation of station snow depth data within operational hydrological models. This remark pertains both to the Introduction and to the Discussion or conclusion parts.

Overall my appreciation of the paper is positive and I encourage its publication provided the above main comment and the following minor comments are addressed.

Thank you for your positive evaluation.

We agree that the discussion on detailed applications is useful. We added following paragraph in the end of Discussion section (Line 419-430):

"It is encouraging that snow depth and SWE distributions are generally well approximated by the Gaussian or weak non-Gaussian distribution, which is a fundamental assumption for statistical characterization of sub-gird variability used in snowpack estimation by remote sensing. The non-Gaussianity found in the partially snow-covered areas may also be modeled by truncated normal distribution although it must be tested further. Moreover, weak non-Gaussian distribution would enable asymptotic methods including the Edgeworth expansion method proposed by Pires and Perdigão (2007). For instance, the non-Gaussian asymptotic method or information metric can effectively determine the saddle point approximation of the joint probability density functions (PDF) through maximizing the Shannon entropy between the remotely sensed signal and the SWE. Thus, the quantification of non-Gaussiany in snow depth/SWE distributions would be an important milestone toward accurate snow water quantification using remote sensing techniques as well as grid-based snow and earth surface models."

**Minor comments**

P2 L44 : you could cite here a bit more literature in support of this statement and extend it to regional climate modelling (for instance citing *Rudisill et al 2024, Lalande et al 2023*)

Thank you. They are nice publications to cite here.

P2 L56 : Luce and Turbonton → Tarboton

Corrected. Thank you.

P3 L 61 : SWE → basin-wide SWE

Thank you.

P3 L83 : I am not a statistician expert, but I would argue that the landing location is affected by microtopography and meteorological effects at the micro-scale (e.g preferential deposition downwind of a crest); is this compatible with "identically distributed"; isn't there a scale effect or spatial aspect to consider?

The phrase, "without microtopography and meteorological effects" was added for accuracy here. Thank you.

P3 L 87-89 : " This implies the presence of both systematic (non-Gaussian) and random (Gaussian) mechanisms in snow accumulation and ablation processes.". I don't see the implication link with the previous sentences. Or rather : I see it, but I think that the meaning of "in theory"

The sentence was revised as follows:
"He, Ohara, and Miller (2019) reported non-Gaussian snow distribution in open areas as well as Gaussian snow distributions in the forested, fully snow-covered areas during the peak snow season using airborne Light Detection and Ranging (LiDAR) observations in the Snowy Range, Wyoming."

L82 should be clarified to make this paragraph clearer (If I understand correctly, all the micro-scale/topographic effects of my previous comment are excluded from the initial "in theory" of the paragraph, but this should be explicitly stated)

I think that the revision above adheres to the context of the paragraph.

P5 L 142 : Hydrology → hydrology

Corrected. Thank you.

P5 L 148 : I understand the interest of having statistics of SWE instead of snow depth for hydrological purposes, but are there other motivations behind the use of SWE instead of Snow depth from ASO data ?

Water resources evaluation purpose was the reason to use SWE data. Snow depth data includes more ineffective records in the dataset.

P5 Fig 1 : the equivalent of Fig 1 for the non-Arctic sites would be great, as well as a table with a short description of the different sites (or sub-sites)'characteristics : extent of the data collection zone, spatial resolution of the data, estimated accuracy, date, collection method (GPR, magnaprobe, etc...), vegetation cover/variability, landform(s).

Table 1 was added. Thank you for nice suggestion.

P6 and further : Some study sites lack a detailed description of topography. For instance in subsection 3.1 too little info is given on this aspect ;

Following sentence was added on Line 176-178:
The topography of these grids in the ACP are very flat with elevation variation of less than 1 meter while accurate absolute elevation data are hard to compare due to the spatial inaccuracy of the magnaprobe.

Fig 3 entails iso-altitude lines but we don't know their altitude spacing ;
They are 1 meter interval contour lines.  The caption now includes, "… superimposed over the land cover map with 1 meter interval contour lines."

line 241 the polygons are mentioned but we learn only at the very end of the manuscript that there are both low-centered and high-centered.
Thank you.  A phrase "found in lower and higher center parts in the left panel of Figure 5" was added on Line 255.

P8 Sect 3.2 : the spatial resolution of the GPR data should be specified for comparison with other monitoring methods
The GPR data are collected based on the frequency of the pulse, which yields very fine or nearly continuous snow depth data.  We used the resampled depth data at 0.5-meter resolution. New Table 1 includes the specifications of the datasets.

Sect 3.4 : In general in this section, the effect of forest vegetation on the Gaussianity could be better highlighted by providing explicitly SWE distributions on forest-covered areas vs on other areas. Also in this section 3.4 and further in the discussion and conclusion, the **effect of scales** should be more emphasized : snow depth/SWE on the forest floor may be quite Gaussian when looked at the spatial scale of over a few meters, but at decimetric or centimetric scales this is likely not true.

L 370 : much stricter → less stricter seems more accurate to me (?)
Stricter is correct.

L377 : I very much like this way of synthesizing your findings.
Thank you!

**References**

Lalande, M., Ménégoz, M., Krinner, G., Ottlé, C., & Cheruy, F. (2023). Improving climate model skill over High Mountain Asia by adapting snow cover parameterization to complex-topography areas. *The Cryosphere*, *17*(12), 5095-5130.

Rudisill, W., Rhoades, A., Xu, Z., & Feldman, D. R. (2024). Are atmospheric models too cold in the mountains? The state of science and insights from the SAIL field campaign. *Bulletin of the American Meteorological Society*.

---

## Author Comment (AC3)

Our responses are denoted in blue color below.

Review of the paper "Characterization of Non-Gaussianity in the Snow Distributions of Various Landscapes" by Ohara et al.

The topic of this paper is interesting. Representing the spatial variability of snow in modeling has been a longstanding challenge, with various approaches proposed by different researchers. However, none of these approaches has proven superior to the others. This paper provides a good test of the idea of using negentropy to evaluate the non-Gaussianity of snow. I recommend accepting the paper with minor revisions. Here are some comments from my perspective.

Thank you for your support on this publication.

General comments:

1. From the snow depth survey using GPR in Inigok (Figure 4), this study mentions that 'the snowdrift due to steep terrain is considered a major source of non-Gaussianity'. We know that the terrain over the Tuolumne River and East River Watersheds varies dramatically, and I would expect strong non-Gaussianity from these watersheds. However, the computed negentropy for fully snow-covered cells in these watersheds was quite small. Could the authors explain why this is different from the conclusion drawn from Figure 4?

It is a good point. Sampling interval (spatial resolution) of data for may be too large to illustrate the wind snowdrift effect while the vegetation effect may reduce the negentropy. We added the following paragraph on Line 321-328.

"Additionally, the spatial resolution of 50 m may be too coarse to capture the local snowdrift effect discussed in sections 3.2 and 3.3. using the very fine resolution data since snowdrift extent around steep cliff is often smaller than the resolution of medium to large scale snow products. Therefore, even with fully snow-covered areas, fine resolution data is required for snowdrift characterization which is potentially important for more accurate snow storage estimation. However, further study is recommended using finer resolution snow data although the combined effect of steep terrain and vegetation on snowdrift is highly complicated and hard to characterize even with modern remote sensing technology."

We added the following paragraph on Line 344-349.

"However, it is interesting that the range of negentropy remains less than 0.5 in fully snow-covered areas in in panel D despite very steep topography in the East River watershed. At Inigok, for example, it is a flat/low-rolling-hills landscape that is punctuated by very abrupt, very steep bluffs that cause the large drifts. In contrast, while East River certainly has much more total topographic relief, it does not have the same long, flat fetch area where the wind can build unimpeeded, nor does it have similar abrupt erosional bluffs."

1. Based on the calculated negentropy, this paper mentions that 'Most of the fully snow-covered areas fell into the category almost Gaussian.' I am curious if this is a conditional conclusion since the paper lacks information on the sensitivity of this index to the spatial scale. For example, the paper uses a 30-meter moving window and a 1500-meter moving window for different datasets. Would such inconsistency be a concern in drawing the conclusion?"

No, the difference in window size (w_size) is not a concern for the conclusions drawn despite the limitation caused by the snow data sample interval discussed above.

The window size was determined by the unbiased estimator of sample statistics. When the window size is too small, the sample estimator of negentropy, which relies on the 3rd and 4th order cumulant (~ moments) estimations, becomes less stable. There is a rule-of-thumb for the sample size n that may be larger than 10^k for reliable k-th order moment estimation (no reference available). As such, we selected 30 points (1500m for ASO snow product, and 30 m for CALM data) for the presentation. The results of second row in Figure 7 with various moving window sizes are shown below:

[Figure]

w_size = 10 (=10x50m = 500m); sample size = $10^2$ = 100 per window

[Figure]

w_size = 20  (20x50m = 1000m); sample size = $20^2$ = 400 per window

w_size = 30  (30x50m = 1500m); sample size = $30^2$ = 900 per window

w_size = 50 (x50m = 2500m); sample size = $50^2$ = 2500 per window

With w_size = 10, the negentropy estimation becomes less reliable and misleading due to the artifacts or error of the estimator.  However, since all the results for various wind sizes conserve the general characteristics, the conclusions drawn are considered effective.

1. I wonder if this paper can include a paragraph in the discussion section to explicitly mention the advantages of using negentropy in describing snow distribution. Otherwise, there are other simple statistical metrics, such as skewness and kurtosis, that can identify non-Gaussianity straightforwardly.

To identify the non-Gaussianity of given samples, skewness may be a simple $3^{rd}$ order measure. However, negentropy is much better statistic based on the Edgeworth expansion as derived in section 2.  The negentropy precisely measures the difference (the Kullback–Leibler divergence) between a given distribution and the Gaussian distribution with the same mean and variance.  The kurtosis ($4^{th}$ order only) does not make sense for this purpose.

Specific comments:

Line 107, Need to explain what is $p_x$.

Px is the sample probability distribution.  It was added to the text (Line 123).